# ODCoNNet: OD Pairs based Conditional Travel Time Inference Network

## Abstract

Estimating travel time in navigation systems enhances user decisions. Current path-based models face high computational costs due to their reliance on path-specific calculation. In this paper, we propose ODConNet(OD Pairs based Conditional Travel Time Inference Network), a model that leverages origin-destination pairs to reduce estimation time. ODConNet employs multi-scale cells to model traffic networks, learning common routes and travel times simultaneously. As a result, the model trained on M(suburb-scale) and S(intersection-scale) cells achieved an inference time of 0.2ms per trajectory, with MAE of 4.3minutes and MAPE of 20.9. This approach enhances the memory efficiency of navigation systems and improves traffic-based search service.

## 1 Introduction

Navigation systems have evolved beyond simple route guidance to provide travel time prediction services, now serving as an essential tool for modern mobility. Travel time predictions offer intuitive information on the estimated arrival at the destination, directly influencing decisions such as schedule management, route selection, and avoidance of traffic congestion. Consequently, travel time inference is considered a critical factor determining user satisfaction in navigation services (Balster et al., 2020).

Travel time inference can be classified into link-based and path-based approaches. Link-based methods predict travel time by estimating the duration for each segment (link) and aggregating these times, resulting in fast computation speeds. However, such methods tend to accumulate errors and provide low accuracy, as they do not account for inter-link correlations, hence failing to capture holistic route characteristics.

To overcome these limitations, path-based prediction techniques have been developed. These leverage historical route data to learn the dependencies between links, allowing for more accurate travel time estimation. Specifically, approaches using Graph Attention Networks (GAT) can predict detailed travel times for individual links, thereby enhancing the efficiency of navigation services (Fang et al., 2021). Additionally, studies employing Graph Neural Networks (GNN) for path-level travel time prediction have also contributed significantly to the advancement of navigation systems (Son et al., 2021).

Although path-based travel time prediction performs well when complete path sequences are available, its applicability becomes limited in scenarios where such information is absent. In practice, the exact route sequence may be difficult to determine or track in real time. For example, when users do not choose the optimal route suggested by the navigation system, or when routes change dynamically due to traffic conditions, models reliant on route sequences may suffer reduced predictive performance. There are also challenges regarding the availability of detailed link information and the computational costs incurred during route sequence generation and processing, which ultimately restrict real-time prediction capabilities.

Recently, there has been growing research on predicting travel time using only origin and destination information. Examples include machine learning approaches that cluster historical, similar origin-destination (OD) pairs to infer travel times (Uber Engineering, 2022), as well as methods that model the correlation structure between links to estimate travel time distributions (Lin et al., 2023). While various studies have been conducted to predict travel time distributions, most still depend on learning

link information, resulting in high computational complexity and prolonged inference times. Uber has addressed link-level complexity by learning connectivity based on coordinates and tiles to ensure real-time operation (Lagos et al., 2024). Furthermore, other studies have explored grid-based input of driving data for travel time prediction.

This study aims to provide rapid and accurate travel time estimates to users by generating cells capable of learning diverse link attributes and designing a network to infer travel time based on these derived cells. The remainder of this paper is organized as follows. Section 2 describes the driving data, cell generation methods, and the architecture of ODConNet (OD Pair based Conditional Travel Time Inference Network) used in our framework. Section 3 presents the analysis and discussion of the accuracy and inference time of the proposed cell generation and ODConNet model. Finally, Section 4 concludes the paper and suggests directions for future research.

## 2 MAIN BODY

### 2.1 DATASETS

In this study, driving trajectory data collected from Seoul and its metropolitan area were utilized to train and evaluate the travel time inference model. The spatial coverage of the dataset is limited to the region including Seoul, encompassing movements within an area of 44.1 km (width) × 29.4 km (height), which corresponds to 9 (horizontal) × 6 (vertical) cells at Geohash level 5. Figure 1 illustrates the spatial range of the dataset used in this study.

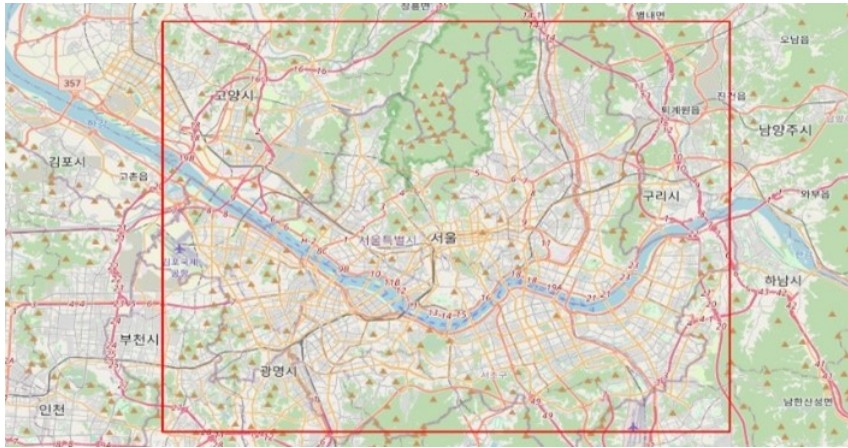

Figure 1: Spatial coverage of the trajectory data

The driving data were collected over a two-week period under normal conditions, excluding special cases such as public holidays, and include a total of 3,565,396 travel routes. Each trajectory $T$ is represented as $T = \{(x_O, y_O, t_O), (x_i, y_i, t_i), \ldots, (x_D, y_D, t_D)\}$ where $(x_i, y_i, t_i)$ denotes an individual trajectory point. Here, $x$ and $y$ represent the longitude and latitude from GPS data, and $t$ indicates the timestamp at each GPS point. $O$ and $D$ refer to the origin and destination, respectively, and $i$ is sampled at an average interval of 14 seconds based on the departure time $t_O$, with the coordinates and time information from each point extracted to construct path $T$.

For travel time inference in this study, origin-destination (OD) pair information was utilized. Each OD pair is defined as $\overline{(OD)} = [x_O, y_O, x_D, y_D, t_O]$ which is used to predict the destination arrival time $t_D$.

The extracted driving data contained outliers—such as records from transport or stationary vehicles—due to diverse driving behaviors, which could adversely affect model training. Therefore, outlier data were filtered out based on the following three criteria: First, data were excluded if the average interval between trajectory records for a route exceeded 80 seconds, as this could not guarantee trajectory continuity. Second, trajectories with a travel distance or straight-line distance between origin and destination less than 0.6 km, or with a travel time of 0 seconds, were considered

abnormal movements and removed. Third, routes whose travel distance was more than twice the straight-line distance between origin and destination were considered inefficient and excluded, to eliminate unrealistic scenarios not reflective of real navigation use cases.

Approximately 70% of the data in Seoul are distributed within a travel time of 8.8–40.8 minutes and travel distances between 2.8 and 19.3 km.

## 2.2 Cell Generation

In this study, the dependence on link-level data was substituted by partitioning the region into a cell-based structure. While link-based learning methods often face challenges with scalability and generalization due to their reliance on road network topology, cell-based approaches were chosen to enhance versatility, as they allow for easy expansion to new regions. Furthermore, collecting sufficient data for each link can be problematic given the vast number of roads, whereas gathering trajectory data at the cell level is more feasible. A key advantage of using cells instead of links is the significant improvement in computational speed, which motivated our adoption of this approach to enhance inference performance.

The road network within a given area comprises a diverse hierarchy, ranging from major express-ways and national roads to smaller local streets and alleyways. The types of roads used vary depend-ing on the area and distance of travel, and this information plays a crucial role in determining travel time. Multi-scale analysis is a technique that represents the same area at various spatial resolutions to simultaneously capture both detailed and broad patterns of urban traffic.

To reflect the hierarchical characteristics of traffic, this study processes data by dividing the same region into cells of three different scales: Large (L), Medium (M), and Small (S). The L-scale cells, representing the largest area, are analogous to administrative districts and are designed to model wide-area traffic flows for long-distance travel. M-scale cells correspond to smaller administrative units ("dong") and capture regional traffic patterns. Finally, the S-scale cells, being the most gran-ular, represent areas roughly the size of an intersection and are used to model fine-grained traffic patterns in urban centers. This multi-scale configuration is intended to comprehensively model traf-fic at various scales, thereby improving the precision and generalization capabilities of travel time inference compared to single-resolution data. The characteristics of each cell scale are summarized in Table 1.

| Scale | Cell Size (km$^2$) | # of Cells (Height $\times$ Width) |
|:-----:|:------------------:|:----------------------------------:|
| L | $5 \times 5$ | $9 \times 6$ |
| M | $2.5 \times 2.5$ | $18 \times 12$ |
| S | $0.6 \times 0.6$ | $72 \times 48$ |

Table 1: Specification of Grid Scales

For model training, each trajectory point $(x_i, y_i, t_i)$ was transformed into a three-channel cell rep-resentation. The resulting set of cells is denoted as $W \times H \times C$, where $W$ and $H$ are the number of cells horizontally and vertically, respectively, and $C$ represents the number of driving features. In this study, $C$ is set to 3, with features corresponding to embedded spatio-temporal information. The final embedded route is denoted as $\overline{T}$, where

$$\overline{T} = \{(x_O, y_O, t_O), (x_i, y_i, t_i), \ldots, (x_D, y_D, t_D)\}.$$

Trajectory information is converted into cells based on each scale ($L$, $M$, and $S$), and the resulting cell-based representation of the route $T$, denoted as $\overline{T}$, is used as input to the proposed ODConNet model. Figure 2 illustrates the cells generated for an identical route. The rows, from top to bottom, correspond to the $L$, $M$, and $S$ scales, while the columns, from left to right, depict the embeddings for location, time, and dwell information, respectively.

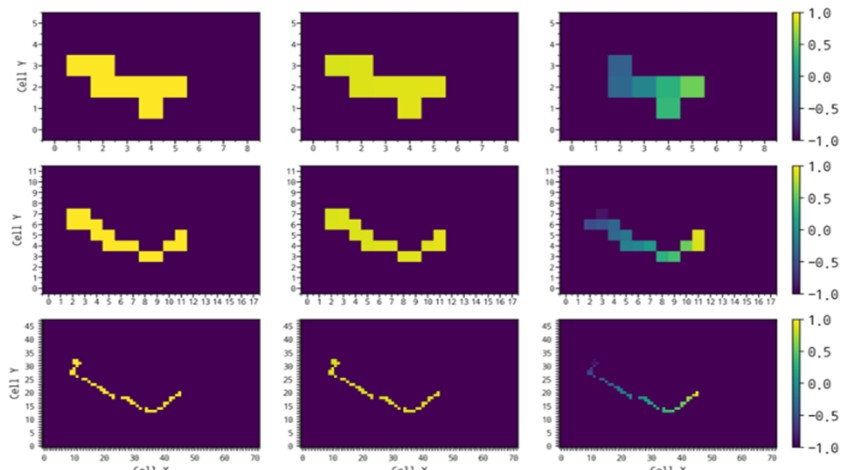

Figure 2: Spatial coverage of the trajectory data

## 2.3 NETWORK ARCHITECTURE

In this study, we developed a network for travel time inference named ODConNet, which is based on a Conditional Variational Autoencoder (CVAE) (Sohn et al., 2015) and the Transformer architecture (Vaswani et al., 2017).

ODConNet is composed of two main components: a training module and an inference module. The training module takes an embedded route, $\overline{T}$, and conditional information, $\overline{(OD)}$, as input, and simultaneously updates the CVAE and Transformer to generate path patterns and learn travel times. The inference module uses the reconstructed route, $\widehat{T}$, from the CVAE and the conditional information, $\overline{(OD)}$, to infer travel time through the Transformer.

The CVAE is conditioned on the OD pair information to precisely generate trajectories that correspond to the given OD pair, capturing the diversity of trajectories through its probabilistic latent space. In our design, the CVAE takes $\overline{T}$ and $\overline{(OD)}$ as input to reconstruct the trajectory. This allows the model to learn the primary route patterns between an origin and destination and to exclude outlier trajectories, thereby enhancing the reliability of the inference. While our CVAE is based on the standard framework, our model is trained using each of the three scales ($L$, $M$, $S$) individually, as well as their combinations.

The Transformer, leveraging its self-attention mechanism, integrates the reconstructed cell sequence, $\widehat{T}$, from the CVAE with the conditional information, $\overline{(OD)}$. This enables it to learn complex spatio-temporal patterns among trajectory elements and perform efficient travel time inference through parallel processing.

The CVAE and Transformer are trained simultaneously through a coupled loss function, $\mathcal{L}_{\text{ODCon}}$. This overall loss function is a weighted sum of the reconstruction loss, $\mathcal{L}_{\text{Recon}}$, the Kullback–Leibler (KL) divergence loss, $\mathcal{L}_{\text{KL}}$, and the Transformer's travel time prediction loss, $\mathcal{L}_{\text{Pred}}$:

$$\mathcal{L}_{\text{ODCon}} = \mathcal{L}_{\text{Recon}} + \mathcal{L}_{\text{KL}} + \mathcal{L}_{\text{Pred}}$$

$\mathcal{L}_{\text{Recon}}$ and $\mathcal{L}_{\text{KL}}$ are derived from the CVAE and are adapted to our trajectory data. $\mathcal{L}_{\text{Recon}}$ calculates the Mean Squared Error (MSE) between the input cells, $\overline{T}$, and the reconstructed cells, $\widehat{T}$. A weight of 1 is assigned to cells containing trajectory information, and a weight of 0.1 is assigned to empty cells, focusing the learning on the actual path taken:

$$\mathcal{L}_{\text{Recon}} = \frac{1}{N} \sum_{i=1}^{N} W_i \left( \overline{T}_i^{(S)} - \widehat{T}_i^{(S)} \right)^2$$

Here, $W$ is the set of per-cell weights, $N$ is the total number of cells for a given scale $S$, and $S$ denotes the cell scale type. The value of $N$ varies by scale: $9 \times 6$ for $L$, $18 \times 12$ for $M$, and $72 \times 48$ for $S$. $\overline{T}_S$ represents the input cell set for scale $S$, and $\widehat{T}_S$ is the reconstructed cell set for scale $S$.

The $\mathcal{L}_{\text{KL}}$ term regularizes the distribution of the latent variables to ensure diversity in the generated trajectories:

$$\mathcal{L}_{\text{KL}} = -\text{KL}\left(q(\mathbf{Z} \mid \overline{T}, \overline{(OD)}) \,\|\, p(\mathbf{Z} \mid \overline{(OD)})\right)$$

The travel time prediction loss, $\mathcal{L}_{\text{Pred}}$, is calculated as the MSE between the predicted and actual travel times:

$$\mathcal{L}_{\text{Pred}} = \frac{1}{M} \sum_{j=1}^{M} (\hat{y}_j - y_j)^2$$

where $M$ is the number of trajectories in the batch, $y_j$ is the ground-truth travel time, and $\hat{y}_j$ is the predicted travel time by the Transformer.

$$L_{\text{Pred}} = \left((t_D - t_O) - (\hat{t}_D - \hat{t}_O)\right)^2$$

This unified loss function concurrently optimizes both the reconstruction quality of the CVAE and the inference accuracy of the Transformer, ensuring consistency between the two models.

When training the CVAE with multi-scale cells, the depth of the convolutional layers was adjusted for each scale to handle varying cell sizes. The S-scale uses four convolutional layers, the M-scale uses three, and the L-scale uses two. This design allows the model to capture complex patterns from high-resolution cells while maintaining computational efficiency at lower resolutions. Each layer incorporates batch normalization to stabilize training and uses the LeakyReLU activation function, which preserves gradients for negative inputs, enabling effective learning of complex trajectory patterns. The decoder has a symmetric structure to the encoder of the same scale, utilizing transposed convolutions.

Figure 3 provides an overview of the ODConNet architecture, illustrating an example where M and S scale cell information is used as input.

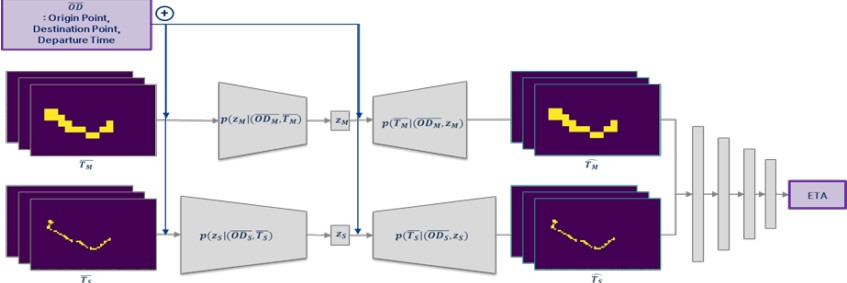

Figure 3: ODConNet Architecture

## 3 RESULTS AND DISCUSSION

### 3.1 CASE ANALYSIS VIA TRAJECTORY VISUALIZATION

In this section, we analyze the model's reconstruction results of trajectory data, highlighting its strengths by visualizing the quality of reconstructed images as heatmaps. Figure 4 presents a visual comparison of the model's trajectory reconstruction and travel time inference performance through two representative cases.

Case 1 demonstrates the model's ability to effectively reconstruct real trajectories, filter out outlier routes, and extract the primary path. Panel (c) accumulates all trajectories between a specific origin-destination (OD) pair, where more frequently traversed paths appear brighter on the heatmap. This visualization reveals both the main routes and infrequent outlier paths, thereby showcasing trajectory diversity. Panel (a) depicts the actual trajectories for the same OD pair, while panel (b) illustrates the model-reconstructed paths for $\overline{(OD)}$, emphasizing reconstruction focused on the most frequent routes. Some reconstructed images include paths deviating from the main route, indicating that the model learns to prioritize key trajectories while still capturing data variability.

Case 2 features an OD pair not included in the training data, where the model exhibited strong reconstruction and travel time inference performance. This scenario involved unusual trajectories and abnormal travel times, making accurate inference challenging. Panel (d) in Figure 4 shows actual abnormal trajectories generated due to waypoints; panel (e) presents the model-reconstructed path for $\overline{(OD)}$, with a red rectangle highlighting the differences between the two. The yellow line in panel (f) indicates the navigation system's recommended route. Notably, despite the actual trajectory being abnormal, the reconstructed trajectory by the model shows high agreement with the recommended path, suggesting that the model leverages learned local knowledge near origins and destinations to reconstruct key routes from anomalous inputs.

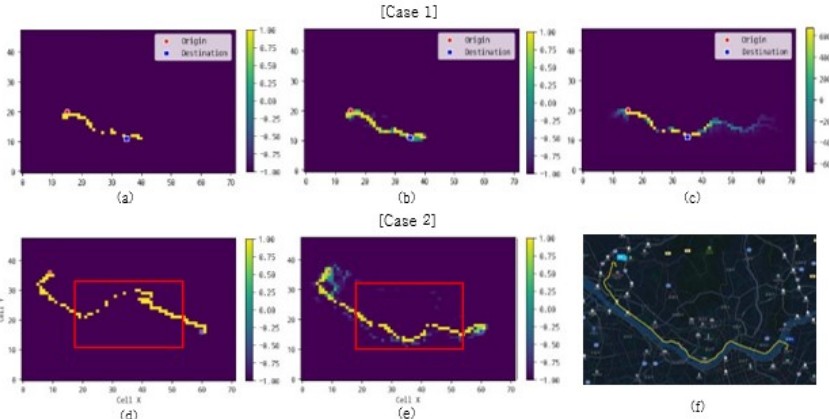

Figure 4: Comparison of actual and reconstructed trajectories: [Case 1] Reconstruction of main routes, [Case 2] Reconstruction from abnormal trajectories

Additionally, the model infers a travel time of 56.9 minutes, which is close to the navigation system's estimate of 61 minutes (within a 6.7% margin of error), thereby demonstrating high accuracy. In contrast, the actual travel time was 109.6 minutes due to abnormal routing with waypoints. This indicates that even for untrained OD pairs, the model reliably infers key routes and travel times based on local contextual information.

Cases 1 and 2 respectively demonstrate the model's proficiency in extracting primary routes for trained OD pairs and its generalization capability for novel OD pairs. These results suggest that the proposed model can be effectively applied to route and travel time inference in navigation systems.

### 3.2 MODEL PERFORMANCE EVALUATION

The proposed model was trained using single scales (L, M, S) and combinations thereof (LM, LS, MS, LMS), and its performance was evaluated on the test dataset, which includes 356,563 trajectories. Evaluation metrics were Mean Absolute Error (MAE) and Mean Absolute Percentage Error (MAPE). Each model variant uses cells from the respective scales as input, and LMS involves the integration of all scales. The results are summarized in Figure 5.

The MS model achieved the best performance, outperforming even the full-scale LMS model. This suggests that the combination of M (medium) and S (small) scales effectively captures both intermediate and detailed traffic patterns, thus enhancing the accuracy of travel time inference. In contrast, the L (large) scale exhibited the lowest performance among single-scale models due to its low spatial resolution, and in LMS, it likely introduced noise rather than providing additional useful information. The S scale recorded the highest performance among single-scale variants, indicating that high-resolution data play a significant role in accurate travel time inference. To verify the effectiveness of the CVAE component, we compared the MS model from this study with a baseline that used only a Transformer, trained on both M and S scale data (without CVAE). As presented in Figure 5, the MS model achieved MAE of 4.26 minutes and a MAPE of 20.9%, while the Transformer-only model showed inferior performance with a MAE of 18.3 minutes and MAPE of 79.9%. This highlights that CVAE improves the accuracy of travel time inference by generating OD pairwise path patterns and reconstructing abnormal trajectories into principal routes.

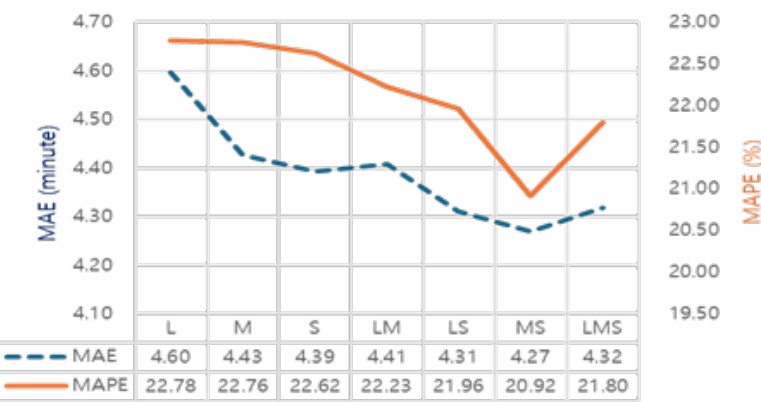

Figure 5: Model performance by scale combinations

## 3.3 TRAVEL TIME INFERENCE RESULTS

This section discusses the travel time inference performance. Figure 6 compares actual average travel times (orange) and the MS model's predicted mean travel times (blue) at 20-minute intervals to evaluate the model's inference capability. The predicted travel times closely follow the actual patterns, demonstrating robustness even during peak congestion periods at 7 AM and 6 PM.

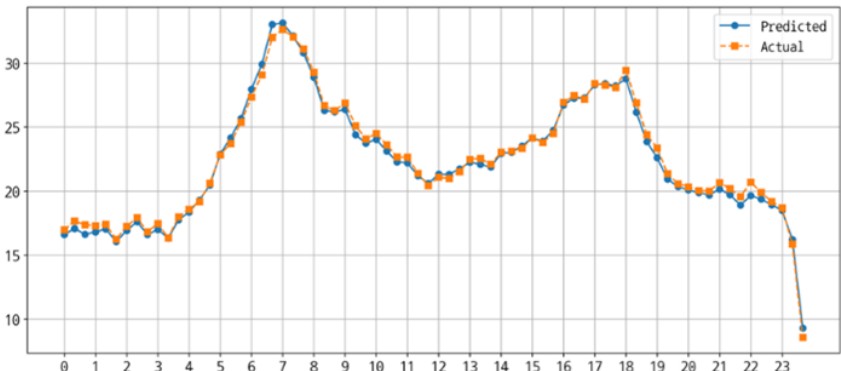

Figure 6: Comparison of actual average travel time and predicted average travel time by time of day

## 3.4 MODEL INFERENCE EFFICIENCY

The proposed model is well-suited for real-time travel time inference in navigation systems. Table 2 compares the inference speed of our MS model against the Hyundai's navigation route search engine and route-sequence-based ETA prediction model, other navigation engines, and the MS model in this study.

The MS model is approximately 2,000 times faster than our route search engine, and 7.5 times faster than our route-sequence-based prediction model. It also significantly outperforms competitor navigation systems: for route searches in downtown Seoul, inference speed was about 2,722 times faster than Naver Maps and 1,735 times faster than Kakao Maps; for routes in the United States, it was about 4,406 times faster than Google Maps. This exceptional speed is attributed to the MS model's use of multi-scale data instead of path-based approaches, and its maximized computational efficiency. These results affirm the MS model's suitability for real-time information provision and its superiority over other existing systems in terms of inference efficiency.

| Category | Model/System | Inference Speed (ms/case) |
|---|---|---|
| Hyundai | Route Search Engine | 400.0 |
| | Route-Sequence-Based Prediction Model | 1.0 |
| Others | Naver Maps | 844.5 |
| | Kakao Maps | 347.0 |
| | Google Maps | 881.2 |
| This Study | MS model | 0.2 |

Table 2: Comparison of Inference Speed across Navigation Systems and Models

## 4 RESULTS AND DISCUSSION

This study proposes a novel deep learning model for travel time inference without route sequence information. The main contributions of this paper are as follows: First, we introduced a unified pipeline model combining the loss functions of CVAE and Transformer, achieving travel time inference using only the origin, destination, and departure time, without route sequence information.

Second, we effectively integrated multiple spatial resolutions by using multi-scale cells. Third, experiments demonstrate that the proposed model surpasses existing route-sequence-based models in terms of computational efficiency.

This research broadens the applicability of travel time inference within navigation systems, especially in environments where route information is restricted. Two main practical implications are anticipated through real-time inference.

From the customer perspective, when searching for destination keywords in navigation, the model can immediately display the estimated arrival times to various candidate destinations, aiding in destination selection.

From the operational system viewpoint, while current mobile app navigation services estimate travel times using route-sequence pattern speed data, adoption of the proposed model eliminates the need for route sequence information, reducing memory usage and enabling faster travel time estimation.

However, this study did not incorporate external factors affecting real-time traffic, such as traffic volume, weather, and destination characteristics. Moreover, since the analysis was conducted using data from Seoul, there may be limitations in the generalizability of the results to other regions. Future research should focus on incorporating additional influential factors into the model to further improve inference performance.

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
