# OpenReview forum: "ODConNet: OD Pairs based Conditional Travel Time Inference Network"
_ICLR.cc/2026/Conference — ICLR 2026 Conference Desk Rejected Submission_

### Official Review · Reviewer_Tuw2 · 2025-10-18

**Soundness:** 1
**Presentation:** 1
**Contribution:** 2
**Rating:** 0
**Confidence:** 5

**Summary:**

The paper proposes a new framework for predicting Origin-Destination time. The framework consists of a Conditional Variational Autoencoder followed by a Transformer model.

**Strengths:**

The Dataste description covers almost all of the processes applied to the data, increasing the reproducibility of the paper. Improve the final phrase by adding the number of data points and any other relevant information.

The description of the training error function.

**Weaknesses:**

The introduction can be improved. The problem is briefly described, and the approach shifts from a graph-based to an independent use of OD data. This creates confusion, and the final problem can not be completely understood until subsection 2.2.

Several sentences can be improved by adding a reference. A sentence without a reference could be considered a claim, and in those cases, the paper should try to demonstrate that, for example, the sentence from lines 042-043. This problem is evident throughout the paper.

The related work must be improved. There are several papers on OD time travel prediction, but the current submission lists only four, being the newest from 2024.

Geohash level must be described.

The paper's problem description is confusing. In subsection 2.1 the paper mentions that OD pair information was utilized. However, later in subsection 2.2, each trajectory is converted to a cell representation at the problem level, indicating that the entire trajectory is needed for the proposed framework.

Regarding Figure 2, what are the ideas of the columns? I understand that these represent the embeddings for location, time, and dwell information, but no explanation is given.

L_{pred} is not used in the paper. Use numbers for the equation and reference them in the text.

The description of the framework must be improved. The lack of information about the proposed framework reduces the reproducibility of the paper. For example, there is no description of the current architectures. As a simple idea, the paper can present the proposed framework in a figure rather than the architecture of the ODConNet.

Several issues call my attention regarding the proposed framework. The transformer model starts with an embedding of the data. Why do you use another convolutional model to extract the information? Also, the original transformer model is for text, but in this case, the model's input is the original data's embeddings. Also, the original transformers use two types of inputs, and this is not explained in the paper (the decoder and encoder).

In general, it seems that the framework is using a vision transformer, and the actual data is the image generated by the cell transformation. If this is the case, then the paper must be rewritten to improve the explanation of this use.

Improve the use of figures. While the figures are referenced in the text, most are basic examples rather than something useful for the paper. For example, Figure 3 shows the architecture of the ODConNet, but the text gives it almost no discussion or importance. Also, several figures have very low resolution or include information not described in the paper (see the scales in Figure 4).

Figure 4 plots e and f seems to be completely different. As a suggestion, the paper should change the subjective results to something objective.

The paper must report the training, validation, and test results to check for overfitting. It should show the error over the number of epochs during training. Actually, the training process is not mentioned in the paper, reducing its reproducibility.

The paper does not use any other baselines to compare. Considering the large number of papers based on this type of data, the paper should compare against other baselines.

Subsection 3.4 does not describe the experiment used to compare fairness.

Minor comments:
-”dong” it should be ``dong”

**Questions:**

What is the input and output of each model?

How does the paper adapt the transformer model for this data?

---

### Official Review · Reviewer_iGRd · 2025-10-28

**Soundness:** 1
**Presentation:** 1
**Contribution:** 1
**Rating:** 2
**Confidence:** 5

**Summary:**

This paper proposes a travel time estimation method for given OD pairs without using trajectory data. The road network is divided into multi-scale cells, and trajectory reconstruction is used to enhance estimation accuracy. However, the paper lacks clear motivation and fails to present any unique technical challenges. The proposed technique is straightforward and offers little innovation or adaptation difficulty for the ETA problem. Moreover, the paper does not include comparisons with baseline methods, making it unsuitable for publication at ICLR.

**Strengths:**

The paper applies the conditional VAE technique to address the OD travel time estimation problem.

**Weaknesses:**

**W1:** The motivation is unclear. The authors are encouraged to review existing OD travel time estimation methods (e.g., [1], [2]), summarize their limitations, and clearly explain how the proposed approach addresses these shortcomings.

**W2:** The paper does not specify the key challenges of the problem, making it unclear why the conditional VAE technique is necessary or appropriate.

**W3:** The description of **ODConNet** lacks sufficient technical detail, and the use of the VAE technique appears straightforward, offering limited methodological novelty for solving the problem.

**W4:** The experimental design is overly simple, with no comparison against baseline methods, making it difficult to evaluate the effectiveness or advantages of the proposed approach.

**W5:** The presentation quality is poor; several figures are unclear and appear to have been plotted without vector (SVG) graphics, reducing readability.


[1] Yaguang Li, Kun Fu, Zheng Wang, Cyrus Shahabi, Jieping Ye, and Yan Liu. *Multi-task Representation Learning for Travel Time Estimation.* SIGKDD, 2018.

[2] Yan Lin, Huaiyu Wan, Jilin Hu, Shengnan Guo, Bin Yang, Youfang Lin, and Christian S. Jensen. *Origin-Destination Travel Time Oracle for Map-Based Services.* SIGMOD, 2023.

**Questions:**

please see the weakness points

---

### Official Review · Reviewer_sCwr · 2025-10-29

**Soundness:** 2
**Presentation:** 1
**Contribution:** 1
**Rating:** 2
**Confidence:** 4

**Summary:**

While the paper presents a practical and efficient approach to travel time estimation using only origin–destination (OD) pairs, it falls short of the novelty, rigor, and generalizability expected for publication at ICLR. The work is more engineering-oriented than foundational, and several methodological and experimental limitations undermine its scientific contribution.

**Strengths:**

Strengths

Clear Motivation: The paper correctly identifies a real-world limitation of path-based travel time prediction, its dependency on full route sequences, and proposes a lightweight alternative that only requires OD pairs and departure time.

Impressive Inference Speed: The reported inference time of 0.2 ms per query is compelling, especially compared to commercial systems. This could be valuable for real-time navigation services.

Multi-Scale Grid Representation: The use of L/M/S spatial grids to capture traffic patterns at different granularities is sensible and empirically validated (MS combination performs best).

CVAE Integration: The CVAE module effectively filters anomalous trajectories and reconstructs plausible routes, as shown in the case studies. Ablation results confirm its importance.

**Weaknesses:**

The architecture combines standard components (CVAE + Transformer) without introducing new modeling ideas. Similar conditional generative frameworks have been widely used in trajectory modeling and time-series forecasting.

Experiments are conducted exclusively on Seoul data over a two-week period, with no cross-city or cross-country validation. This raises serious concerns about generalizability to other urban structures, traffic regimes, or driving cultures.

The paper compares inference speed against commercial APIs but does not compare accuracy against state-of-the-art academic models like DeepETA, OD Oracle on the same dataset.
Without such comparisons, claims of “accuracy” (MAE = 4.26 min, MAPE = 20.9%) lack contex and are these numbers competitive?

The model ignores real-time traffic conditions, weather, road types, POI semantics, and historical congestion patterns—all standard inputs in modern ETA systems. While the authors acknowledge this, the omission significantly limits practical applicability.

Even static road network features (e.g., speed limits, lane count) are absent, despite being publicly available and highly predictive.

Reproducibility Concerns:
Key implementation details are missing: How are grid sequences fed into the Transformer? Is positional encoding used? How are M and S scales fused (concatenation, attention, etc.)?
The loss weighting scheme (1 for occupied cells, 0.1 for empty) is heuristic and not justified.

**Questions:**

See Weaknesses.

---

### Official Review · Reviewer_LfrR · 2025-11-02

**Soundness:** 2
**Presentation:** 2
**Contribution:** 2
**Rating:** 2
**Confidence:** 3

**Summary:**

The paper introduces ODConNet, a deep learning model for travel time inference that operates using only origin-destination (OD) pairs and departure time, without requiring detailed route (path) sequences. The proposed approach combines a Conditional Variational Autoencoder (CVAE) to reconstruct routes between OD pairs, and a Transformer network to predict travel time conditioned on reconstructed trajectories.
To model spatial structure, the city is divided into multi-scale cells , Large (L), Medium (M), and Small (S, representing progressively finer geographic granularity. They performed experiments on a dataset of 3.56 million trajectories from the Seoul metropolitan area.

**Strengths:**

1. The topic is relevant to the transportation based research.
2. The architecture is explained clearly with necessary loss functions.
3. The hierarchical cell-based encoding is an elegant idea to capture both spatial dependencies without requiring explicit graph topology.

**Weaknesses:**

1. No formal justification for why this combination is optimal for OD-based inference. Alongside, there is no comparison with existing baselines related to spatiotemporal CNNs or DeepETA [i].
2. The results lack temporal and geographical diversity as it has been focussed on Seoul for a period of two-week under normal conditions.
3. The paper assumes that route inference using OD suffices ETA prediction, but it ignores multimodality involved in real-time traffic like pedestrian detection, hazard stop etc.



[i] Wu, F., & Wu, L. (2019). DeepETA: A Spatial-Temporal Sequential Neural Network Model for Estimating Time of Arrival in Package Delivery System. Proceedings of the AAAI Conference on Artificial Intelligence

**Questions:**

Same as weaknesses.

1. What's the reason of such high difference between other navigation systems and the propsed model? How is it measured for the other systems -- is not at all clear.

---

### Note · Program_Chairs · 2026-01-17
**Submission Desk Rejected by Program Chairs**

The following references in this submission do not refer to real documents and/or have major errors in bibliographic information:

 Sangkyu Son, Ahyoung Kang, Ruda Lee, and Jaehong Eom. Development of a graph-based driving speed prediction model using spatiotemporal data. Proceedings of the Korea Computer Congress (KCC), pp. 756-758, 2021.